# Diagnostic Value of Neutrophil-to-Lymphocyte, Platelet-to-Lymphocyte, and Monocyte-to-Lymphocyte Ratios for the Assessment of Rheumatoid Arthritis in Patients with Undifferentiated Inflammatory Arthritis

**DOI:** 10.3390/diagnostics12071702

**Published:** 2022-07-13

**Authors:** Byung-Wook Song, A-Ran Kim, Yun-Kyung Kim, Geun-Tae Kim, Eun-Young Ahn, Min-Wook So, Seung-Geun Lee

**Affiliations:** 1Division of Rheumatology, Department of Internal Medicine, Pusan National University Hospital, Pusan National University School of Medicine, Busan 50612, Korea; medicaldiver@naver.com (B.-W.S.); solees17@naver.com (A.-R.K.); 2Biomedical Research Institute, Pusan National University Hospital, Busan 49241, Korea; 3Division of Rheumatology, Department of Internal Medicine, Kosin University College of Medicine, Busan 49267, Korea; efmsungmo@hanmail.net (Y.-K.K.); gtah@hanmail.net (G.-T.K.); 4Division of Rheumatology, Department of Internal Medicine, Pusan National University Yangsan Hospital, Yangsan 50612, Korea; ahnmolla@hanmail.net (E.-Y.A.); thalsdnrso@naver.com (M.-W.S.)

**Keywords:** arthritis, rheumatoid arthritis, diagnosis, blood cells, inflammation

## Abstract

Background: To investigate the diagnostic performance of the neutrophil-to-lymphocyte ratio (NLR), platelet-to-lymphocyte ratio (PLR), and monocyte-to-lymphocyte ratio (MLR) in the diagnosis of rheumatoid arthritis (RA) in subjects with undifferentiated inflammatory arthritis (UIA). Methods: This retrospective cohort study investigated 201 female patients with UIA (≥1 swollen joint) and 280 age-matched, healthy female controls. “Clinical RA” was defined based on the clinical judgment of a rheumatologist and “disease-modifying anti-rheumatic drugs (DMARDs) RA” was defined as a case of initiating DMARDs treatment within 6 months after the first visit. “Classified RA” was defined as fulfilling the 2010 classification criteria for RA. Receiver operating characteristics were used to determine the optimal cut-off value. Results: UIA patients had a significantly higher NLR, PLR, and MLR than the controls. Among the 201 UIA patients, 65 (32.3%), 63 (31.3%), and 61 (30.3%) subjects were classified as clinical RA, DMARDs RA, and classified RA, respectively. At a cut-off of 0.24, MLR showed moderate accuracy for the diagnosis of DMARDs RA (sensitivity, 65.1%; specificity, 62.3%; area under the curve [AUC], 0.701; *p* < 0.001). However, the diagnostic accuracies of NLR and PLR were low. Conclusions: MLR may be used as a complementary diagnostic indicator for RA diagnosis in patients with UIA.

## 1. Introduction

Rheumatoid arthritis (RA) is a chronic systemic autoimmune disease of unknown etiology characterized by symmetrical peripheral inflammatory synovitis and extra-articular manifestations, such as rheumatoid nodules, interstitial lung disease, and osteoporosis [1,2]. Synovitis in RA can induce progressive irreversible joint damage and deformity leading to functional disability and increased morbidity and mortality [1,2]. Given the introduction of disease-modifying anti-rheumatic drugs (DMARDs), such as methotrexate, and the advent of new highly efficacious biological agents over the last decades, significant advances have been made in the management of RA [3]. Because earlier treatment with DMARDs can contribute to a more favorable clinical outcome [4], the early evaluation and diagnosis of RA is crucial and is considered as an overarching principle in its management [5]. However, there is no single test that can confirm RA, and its diagnosis is primarily established based on the clinical symptoms and serological and imaging tests. As such, its early diagnosis is difficult and frequently delayed. RA usually progresses from an asymptomatic stage to an undifferentiated inflammatory arthritis stage with disease progression. Notably, distinguishing RA from other forms of arthritis in the early stages is imperative in delaying the progression of RA [6].

The neutrophil-to-lymphocyte ratio (NLR), platelet-to-lymphocyte ratio (PLR), and monocyte-to-lymphocyte ratio (MLR), which are calculated from the complete blood count (CBC), are recognized as biomarkers representing systemic inflammation and balance of the immune response [7,8,9]. As the CBC test is inexpensive and routinely performed in most clinical departments regardless of the type of disease, these hematologic markers have the advantages of availability, accessibility, and cost-effectiveness. Recently, a growing interest has been raised regarding the clinical implications of NLR, PLR, and MLR in various diseases, including chronic inflammatory diseases, malignancies, and cardiometabolic diseases. These markers have been reported to not only be associated with disease activity and treatment response in chronic autoimmune diseases, including RA [7,8,10,11,12,13], but also have a prognostic significance in various cancers [14,15,16,17,18] and cardiovascular diseases [19]. However, the diagnostic roles of NLR, PLR, and MLR in RA have not been well studied. Here, we aim to investigate the diagnostic performance of NLR, PLR, and MLR in the diagnosis of RA in female patients with undifferentiated inflammatory arthritis (UIA) in a real clinical setting.

## 2. Materials and Methods

### 2.1. Study Design and Population

This was a retrospective cohort study conducted in the department of rheumatology in a tertiary referral center in Korea. We investigated 201 female patients with UIA aged ≥ 20 years and age-matched (±2 years) female healthy controls who had CBC results between January 2018 and March 2020. Because a previous study reported that NLR levels were higher in male patients with systemic autoimmune rheumatic diseases than in their female counterparts [20], we considered that the selection of only one gender group was appropriate for evaluating the clinical significance of NLR, PLR, and MLR. Additionally, because the number of male patients with UIA was limited, we only analyzed female subjects in this study. UIA was defined as a case presenting ≥1 swollen joint, which could not be explained by another disease. UIA should be regarded as a diagnosis of exclusion and major classes of disease to consider for the differential diagnosis for UIA includes RA as well as osteoarthritis, spondyloarthritis, crystal arthropathies, and connective tissue diseases, such as systemic lupus erythematosus. The following patients with UIA were excluded from the study: (1) patients aged < 20 years; (2) male patients; (3) patients with a history of or have a concomitant hematologic disease, malignancy, active infection, or thrombotic disorders; (4) those who had received DMARDs for the treatment of RA within 6 months before the first visit (index date); and (5) those receiving non-steroidal anti-inflammatory drugs or glucocorticoids within 6 months before the index date. Healthy controls were randomly selected from female patients who visited the health promotion center of the same hospital for comprehensive routine health check-ups and had no history of rheumatologic disease, hematologic disease, malignancy, active infection, thrombotic disorders, diabetes mellitus, hypertension, or chronic kidney disease. Healthy controls did not have any joint-related symptoms, such as arthralgia. The Research and Ethical Review Board of Pusan National University Hospital approved this study and waived the need for informed consent owing to the retrospective study design (IRB no. 2012-020-098).

### 2.2. Clinical Variables

Data regarding age, CBC, and serum C-reactive protein (CRP) levels at the index date were obtained in both patients with UIA and healthy controls. NLR, PLR, and MLR were determined by dividing the absolute neutrophil count by the absolute lymphocyte count, the absolute platelet count by the absolute lymphocyte count, and the absolute monocyte count by the absolute lymphocyte count, respectively.

For patients with UIA, the following variables at the index date were extracted from the medical records: erythrocyte sedimentation rate (ESR), number and location of swollen and tender joints, symptom duration, rheumatoid factor (RF), and anti-cyclic citrullinated protein (CCP) antibody. The titers of RF and anti-CCP antibodies were determined using a particle-enhanced immunoturbidimetric assay (range 0–14 IU/mL) and a chemiluminescent microparticle immunoassay (range 0–5 U/mL), respectively. The use of DMARDs, such as methotrexate, sulfasalazine, hydroxychloroquine, and leflunomide, within 6 months after the index date was investigated. All UIA patients were evaluated using the 2010 American College of Rheumatology/European League against Rheumatism (ACR/EULAR) classification criteria for RA [21], and the criteria score was measured for each patient at the index date. “Clinical RA” was defined based on the clinical judgment of a rheumatologist, regardless of whether or not DMARD therapy was used at the index date. “DMARDs RA” was defined as a case of initiating DMARDs treatment within 6 months after the index date. The decision to administer DMARD therapy was performed by experienced rheumatologists in our center. “Classified RA” was defined as fulfilling the 2010 ACR/EULAR classification criteria for RA at the index date.

### 2.3. Statistical Analyses

Data are expressed as mean ± standard deviation (SD) or as median (interquartile range [IQR]) for continuous variables, and as number (%) for categorical variables, as appropriate. The Kolmogorov–Smirnov test was used to examine the normality of the continuous variables. Continuous data were compared using the Student’s *t*-test or Mann–Whitney U test, and categorical variables were compared using the chi-squared test or Fisher’s exact test, as appropriate. Spearman’s correlation analysis was used to assess the correlation of NLR, PLR, and MLR with other clinical variables. Receiver operating characteristic (ROC) analysis with area under the curve (AUC) was used to determine the optimal cut-off values of NLR, PLR, and MLR to maximize sensitivity and specificity in the diagnosis of clinical, DMARDs, and classified RA. An AUC of >0.9, AUC between 0.7 and 0.9, and AUC between 0.5 and 0.7 were considered to indicate high, moderate, and low diagnostic accuracies, respectively [22]. The sensitivity, specificity, positive predictive value (PPV), and negative predictive value (NPV) were also calculated. All statistical analyses were performed using STATA 15.0 (StataCorp LP, College Station, TX, USA) by A.-R.K, and *p* < 0.05 was considered statistically significant.

## 3. Results

The baseline clinical and laboratory characteristics of patients in the UIA and healthy controls are summarized in Table 1. The median NLR, PLR, MLR, and CRP levels were significantly higher in patients with UIA than in healthy controls. The median symptom duration of UIA patients was 12 months, and the frequencies of RF and anti-CCP antibody positivity were 34.3% and 27.4%, respectively. A total of 65 (32.3%), 63 (31.3%), and 61 (30.3%) patients with UIA were diagnosed with clinical RA, DMARDs RA, and classified RA, respectively. In patients with DMARDs RA, the frequencies of methotrexate, sulfasalazine, hydroxychloroquine, and leflunomide use were 79.4%, 47.6%, 11.1%, and 12.7%, respectively. The frequencies of RF positivity and anti-CCP antibody positivity were 70.8% and 76.9% in patients with clinical RA, 71.4% and 77.8% in patients with DMARDs RA, and 85.2% and 80.3% in patients with classified RA, respectively. A Venn diagram showing the number of clinical RA, DMARDs RA, and classified RA is presented in Figure 1. The majority of RA patients (n = 53) fulfilled all three criteria for RA, whereas 8 patients with RA met only one of the three criteria (clinical RA = 1; classified RA = 7).

Comparisons of NLR, PLR, and MLR among healthy controls, non-RA UIA patients, and RA patients are shown in Figure 2. Regardless of the type of RA, both non-RA UIA patients and RA patients had a significantly higher NLR, PLR, and MLR than healthy controls. The median NLR and MLR levels in patients with clinical RA, DMARDs RA, and classified RA were significantly higher than those without clinical RA, DMARDs RA, and classified RA, respectively. Patients with clinical RA and DMARDs RA had a significantly higher PLR than those without clinical RA and DMARDs RA, respectively. There was no significant difference in PLR between patients with and without classified RA.

Comparisons of clinical characteristics according to the presence of clinical RA, DMARDs RA, and classified RA are described in Appendix A. In addition, the median ESR, CRP, swollen joint count (SJC), total joint count, and frequency of RF positivity and anti-CCP antibody positivity in patients with clinical RA, DMARDs RA, and classified RA were significantly higher than those with non-clinical RA, non-DMARD RA, and non-classified RA, respectively.

The correlation of NLR, PLR, and MLR with other clinical and laboratory variables is shown in Table 2. NLR, PLR, and MLR showed significant positive correlations with SJC, ESR, CRP, and titer of anti-CCP antibody in patients with UIA. In addition, RF titers were positively correlated with NLR and MLR levels. NLR and MLR were significantly positively correlated with the 2010 ACR/EULAR classification criteria score (ρ = 0.208, *p* = 0.003 in NLR and ρ = 0.313, *p* < 0.001 in MLR).

The cut-off values of NLR, PLR, and MLR and their sensitivity, specificity, PPV, and NPV in the diagnosis of clinical RA, DMARDs RA, and classified RA in patients with UIA are described in Table 3. The optimal cut-off values of NLR, PLR, and MLR were 2.07, 143.26, and 0.24, respectively, for the diagnosis of clinical RA, DMARDs RA, and classified RA. At a cut-off of 0.24, MLR showed moderate accuracy for the diagnosis of RA with DMARDs (sensitivity, 65.1%; specificity, 62.3%; AUC, 0.701; *p* < 0.001), but it had low accuracy for the diagnosis of clinical RA (sensitivity, 64.6%; specificity, 62.5%; AUC, 0.687; *p* < 0.001) and classified RA (sensitivity, 62.3%; specificity, 60.7%; AUC, 0.663; *p* < 0.001). Otherwise, NLR and PLR had a low diagnostic accuracy (AUC < 0.7) for the diagnosis of clinical RA, DMARDs RA, and classified RA (Table 3).

Table 4 shows the probability of clinical RA, DMARDs RA, and classified RA in patients with UIA according to the optimal cut-off values of NLR, PLR, and MLR. The probabilities of clinical RA, DMARDs RA, and RA in UIA patients with a cut-off value or higher for all NLR, PLR, and MLR were 46.1%, 44.6% and 45.9%, respectively. Otherwise, the frequencies of clinical RA, DMARDs RA, and RA in UIA patients with less than a cut-off value for all three indicators were 23.8%, 21.5% and 23.1%, respectively.

## 4. Discussion

In this retrospective cohort study, we investigated the clinical utility of NLR, PLR, and MLR for the diagnosis of patients with UIA in a real clinical practice. Patients with UIA had significantly higher NLR, PLR, and MLR levels than healthy controls. In addition, for UIA patients, NLR and MLR levels in patients with all three types of RA (i.e., clinical RA, DMARDs RA, and classified RA) were significantly higher than those in patients without RA. NLR, PLR, and MLR were significantly correlated with the SJC, RA-specific auto-antibody (anti-CCP antibody), and the level of acute phase reactants (ESR and CRP), all of which are important clinical parameters in the diagnosis of RA. Although the overall diagnostic accuracy of NLR and PLR for the detection of RA was low, the MLR demonstrated moderate diagnostic accuracy for DMARDs RA. This result indicates that MLR may be used as a complementary diagnostic indicator for the diagnosis of RA in patients with UIA. Otherwise, approximately half of the patients with RA had NLR, PLR, and MLR above the optimal cut-off value.

To the best of our knowledge, this is the first study to evaluate the diagnostic accuracy of NLR, PLR, and MLR in distinguishing RA from non-RA in patients with UIA. The diagnostic value of these hematologic markers in RA has also been reported previously, but all previous studies investigating this topic focused on the diagnostic role of NLR, PLR, and MLR in discriminating RA between healthy subjects rather than discriminating RA between non-RA in UIA patients [23,24,25,26]. Because clinicians usually encounter patients with joint symptoms rather than healthy, asymptomatic individuals, we believe that our study better reflects the actual clinical situation. In discriminating RA from healthy subjects, the AUC of NLR, PLR, and MLR varied considerably in previous studies, ranging from less than 0.55 to 0.831 [23,24,25,26]. In our study, the AUC value of MLR was 0.701 for detecting RA in patients with UIA. Taken together, although the overall diagnostic accuracy of these hematologic markers was low to moderate, we suggest that MLR could be used as an auxiliary tool for diagnosing RA because it is not only routinely performed at the clinic, but also inexpensive and affordable.

Although the association of NLR, PLR, and MLR with disease activity, prognosis, and monitoring in RA has been extensively investigated [7,27,28], little data regarding the clinical significance of these hematologic indicators in patients with UIA are available. In our data, NLR, PLR, and MLR showed a significant positive correlation with not only inflammatory parameters, such as ESR, CRP, and SJC, but also with the titer of RA-specific autoantibodies, such as anti-CCP antibody in patients with UIA. In addition, NLR, PLR, and MLR in patients with UIA were significantly higher than those in controls. Similar to our results, these hematologic markers also correlated with disease activity and the titer of RF in patients with RA [7,24,26,27,28,29]. Thus, NLR, PLR, and MLR are considered useful biomarkers for disease monitoring in patients with inflammatory arthritis, considering that these markers could reflect both inflammatory burden and the status of autoimmunity.

Neutrophils, platelets, monocytes/macrophages, and lymphocytes are known to be actively involved in the pathogenesis of RA [30,31,32,33,34,35]. Monocytes circulate in the bloodstream and migrate into the inflamed synovium, where they can differentiate into macrophages. RF and anti-citrullinated protein antibodies (ACPA) in RA patients can form immune complexes with citrullinated proteins in the synovium and subsequently activate synovial macrophages, resulting in the production of proinflammatory cytokines, such as tumor necrosis factor-α (TNF-α) and interleukin-6, which are the central processes in the pathogenesis of RA [33]. Activated macrophages activate fibroblast-like synoviocytes (FLS) to promote the secretion of pathogenic mediators, including matrix metalloproteinase and granulocyte macrophage colony-stimulating factors, which cause bone erosion in RA [33]. Neutrophils are the most abundant type of white blood cells and are responsible for the first cellular response to acute inflammation or injury. In the RA synovium, neutrophils not only activate synovial FLS, but also produce TNF-α and receptor activator of nuclear factor kappa-B ligand, which play an important role in the development of synovitis and bone erosion [31]. In addition, RF and ACPA can induce synovial neutrophils to promote the formation of neutrophil extracellular traps, which can contribute to the development of a pathogenic immune response in RA [31]. Although the primary role of platelets is hemostasis and thrombosis formation, they are also actively involved in immune responses by regulating their own inflammatory mediators [32]. The role of platelets in the pathogenesis of RA has not been well elucidated. However, recent studies have suggested that dysregulated platelet activation ligands and pro-inflammatory molecules can result in the activation of platelet signaling pathway, which, in turn, promotes the production of localized chemokines, cytokines, and growth factors, subsequently leading to the exacerbation of synovitis in RA [32]. Otherwise, lymphopenia may progress with the progression of RA because the localized accumulation of lymphocytes in joints might cause a gradual decrease in lymphocyte count [29]. Among these blood cells, monocytes/macrophages may play the most important role in the development and progression of synovitis in RA. Thus, we suggest that MLR has a better diagnostic accuracy for RA in patients with UIA in our data.

There are some limitations to the present study that should be noted. First, because this was a single-center retrospective cohort study, selection bias may be inevitable. Thus, further multicenter prospective studies are needed to confirm our results. Second, the present study could not fully adjust the effect of medications on the levels of NLR, PLR, and MLR due to its retrospective nature, although we excluded UIA patients who received DMARDs, non-steroidal anti-inflammatory drugs, or glucocorticoids at the index date. Third, as we only analyzed female patients with UIA, due to the limited number of male patients with UIA in our center, as mentioned above. Hence, further research is needed to determine the diagnostic accuracy of NLR, PLR, and MLR in their male counterparts in order to make these markers more reliable diagnostic indicators for RA in patients with UIA. Fourth, we only measured the NLR, PLR, and MLR at the index date. Because NLR, PLR, and MLR are affected by numerous variables, longitudinal changes in these markers may provide more information regarding their diagnostic accuracy for the detection of RA in patients with UIA.

## 5. Conclusions

In conclusion, the present study revealed that MLR had a moderate diagnostic accuracy for distinguishing RA from non-RA in patients with UIA, although the NLR and PLR were of limited value for RA diagnosis. Because the CBC test is easily accessible, inexpensive, and reliable, we concluded that MLR may serve as an auxiliary diagnostic tool in the diagnosis of RA in patients with UIA in real clinical settings, especially in primary care clinics. Our data also indicate that NLR, PLR, and MLR reflect not only the degree of inflammation, but also the status of RA-related autoantibodies in UIA. The results of the present study may provide additional insight into the clinical significance of NLR, PLR, and MLR in UIA. However, due to the retrospective nature of our study and the small sample size, further investigations are warranted to validate our findings and determine the effect of blood cells on the progression from UIA to RA.

## Figures and Tables

**Figure 1 diagnostics-12-01702-f001:**
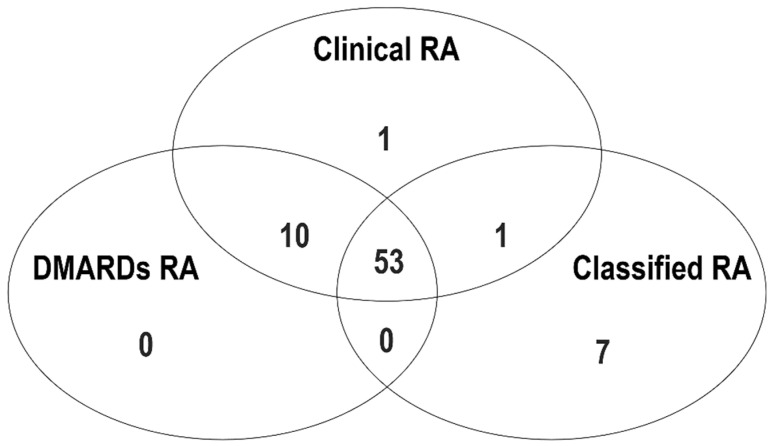
Proportion of “clinical RA”, “DMARDs RA”, and “classified RA” among patients with early inflammatory arthritis.

**Figure 2 diagnostics-12-01702-f002:**
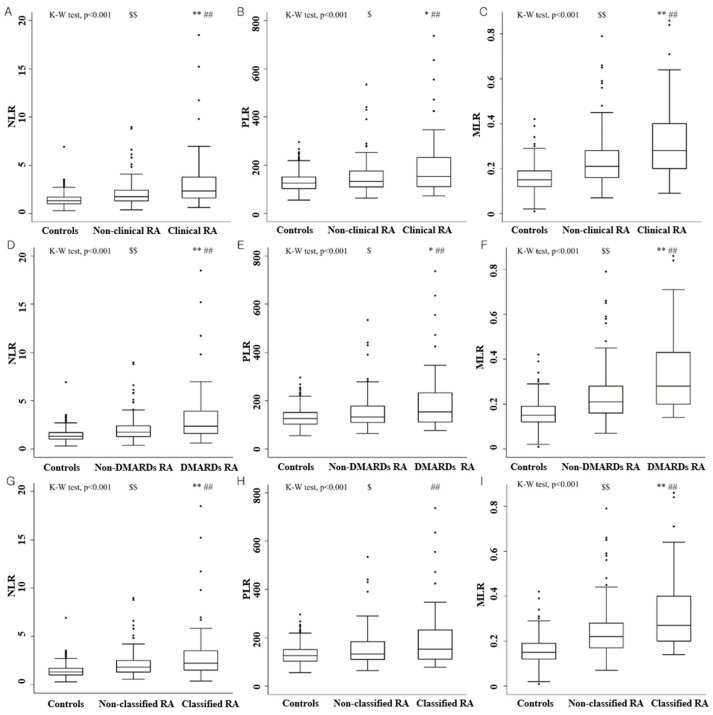
Comparisons of neutrophil-to-lymphocyte (**A**,**D**,**G**), platelet-to-lymphocyte (**B**,**E**,**H**), and monocyte-to-lymphocyte ratios (**C**,**F**,**I**) among healthy controls, early inflammatory arthritis patients without rheumatoid arthritis, and those with rheumatoid arthritis. K–W test: Kruskal–Wallis test, DMARDs: disease-modifying anti-rheumatic drugs, and RA: rheumatoid arthritis. * *p* < 0.05 vs. non-RA, ** *p* < 0.001 vs. non-RA, ## *p* < 0.001 vs. healthy controls, $ *p* < 0.05 vs. healthy controls, $$ *p* < 0.001 vs. healthy controls.

**Table 1 diagnostics-12-01702-t001:** Baseline clinical characteristics between patients with early inflammatory arthritis and healthy controls.

	UIA (n = 201)	Controls (n = 280)	*p* Value
Age, years, mean ± SD	58.8 ± 12.9	58 ± 8.2	0.621
WBC, 10^3^/uL, median, (IQR)	6.46 (5.00–8.14)	5.01 (4.24–5.83)	<0.001
Platelet, 10^6^/uL, median, (IQR)	265 (230–324)	239 (216–277)	<0.001
NLR, median, (IQR)	1.96 (1.34–2.69)	1.32 (1.01–1.70)	<0.001
PLR, median, (IQR)	135.54 (110.1–188.2)	126.18 (102.99–151.2)	<0.001
MLR, median, (IQR)	0.23 (0.18–0.32)	0.15 (0.12–0.19)	<0.001
CRP, mg/dL, median, (IQR)	0.08 (0.03–0.62)	0.03 (0.02–0.07)	<0.001
ESR, mm/h, median, (IQR)	16 (7–34)		
Sx duration, month, median, (IQR)	12 (5–24)		
SJC, n, median, (IQR)	1 (1–2)		
TJC, n, median, (IQR)	2 (1–4)		
Small-joint involvement, n, median, (IQR)	2 (1–2)		
Large-joint involvement, n, median, (IQR)	0 (0–1)		
RF, IU/mL, median, (IQR)	9.5 (7.0–23.2)		
RF positivity, n (%)	69 (34.3)		
Anti-CCP, U/mL, median, (IQR)	28.4 (0.2–10.8)		
Anti-CCP positivity, n (%)	55 (27.4)		
2010 ACR/EULAR criteria score, median, (IQR)	4.0 (3–6)		
Clinical RA, n (%)	65 (32.3)		
DMARDs RA, n (%)	63 (31.3)		
Classified RA, n (%)	61 (30.3)		

UIA: undifferentiated inflammatory arthritis, SD: standard deviation, IQR: interquartile range, WBC: white blood cell, NLR: neutrophil-to-lymphocyte ratio, PLR: platelet-to-lymphocyte ratio, MLR: monocyte-to-lymphocyte ratio, CRP: C-reactive protein, ESR: erythrocyte sedimentation rate, Sx: symptom, SJC: swollen joint count, TJC: tender joint count, RF: rheumatoid factor, CCP: cyclic citrullinated protein, RA: rheumatoid arthritis, DMARDs: disease-modifying anti-rheumatic drugs.

**Table 2 diagnostics-12-01702-t002:** Correlation between laboratory and clinical variables in postmenopausal patients with undifferentiated inflammatory arthritis.

Parameter		SJC	TJC	Sx Duration, Month	ESR, mm/h	CRP, mg/dL	RF, IU/mL	Anti-CCP Ab, U/mL	Criteria Score
NLR	ρ *p*-value	0.251 <0.001	0.123 0.081	−0.14 0.047	0.395 <0.001	0.513 <0.001	0.197 0.005	0.229 0.001	0.208 0.003
PLR	ρ *p*-value	0.181 0.01	0.055 0.441	−0.114 0.108	0.231 <0.001	0.332 <0.001	0.124 0.079	0.164 0.02	0.138 0.05
MLR	ρ *p*-value	0.295 <0.001	0.229 0.001	−0.036 0.61	0.329 <0.001	0.456 <0.001	0.241 0.001	0.239 0.001	0.313 <0.001

SJC: swollen joint count, TJC: tender joint count, Sx: symptoms, ESR: erythrocyte sedimentation rate, CRP: C-reactive protein, CCP: cyclic citrullinated protein, NLR: neutrophil-to-lymphocyte ratio, PLR: platelet-to-lymphocyte ratio, MLR: monocyte-to-lymphocyte ratio.

**Table 3 diagnostics-12-01702-t003:** Optimal cut-off value of neutrophil-to-lymphocyte, platelet-to-lymphocyte, and monocyte-to-lymphocyte ratios to maximize sensitivity and specificity in the diagnosis of clinical, disease-modifying anti-rheumatic drugs, and classified rheumatoid arthritis in patients with undifferentiated inflammatory arthritis.

Diagnosis	Parameter	Cut-Off	Sensitivity	Specificity	PPV	NPV	AUC	*p* Value
Clinical RA	NLR	2.07	66.2%	64%	46.7%	79.8%	0.661	<0.001
	PLR	143.26	56.9%	61.8%	41.6%	75%	0.593	0.034
	MLR	0.24	64.6%	62.5%	45.7%	78.7%	0.687	<0.001
DMARDs RA	NLR	2.07	66.7%	63.8%	45.7%	80.7%	0.671	<0.001
	PLR	143.26	57.1%	61.6%	40.4%	75.9%	0.600	0.023
	MLR	0.24	65.1%	62.3%	44.1%	79.6%	0.701	<0.001
Classified RA	NLR	2.07	62.3%	61.4%	41.3%	78.9%	0.622	0.006
	PLR	143.26	55.7%	60.7%	38.2%	75.9%	0.580	0.070
	MLR	0.24	62.3%	60.7%	40.9%	78.7%	0.663	<0.001

PPV: positive predictive value, NPV: negative predictive value, AUC: area under the curve, DMARDs: disease-modifying anti-rheumatic drugs, RA: rheumatoid arthritis, NLR: neutrophil-to-lymphocyte ratio, PLR: platelet-to-lymphocyte ratio, MLR: monocyte-to-lymphocyte ratio.

**Table 4 diagnostics-12-01702-t004:** Proportion of patient with clinical, disease-modifying anti-rheumatic drugs, and classified rheumatoid arthritis among patients with undifferentiated inflammatory arthritis according to the cut-off values of neutrophil-to-lymphocyte, platelet-to-lymphocyte, and monocyte-to-lymphocyte ratios.

		Clinical RA (n = 65)		DMARDs RA (n = 63)		Classified RA (n = 61)	
		NLR ≥ 2.07	NLR < 2.07	NLR ≥ 2.07	NLR < 2.07	NLR ≥ 2.07	NLR < 2.07
MLR ≥ 0.24	PLR ≥ 143.26	30 (46.1)	2 (3.1)	29 (44.6)	2 (3.1)	28 (45.9)	1 (1.5)
	PLR < 143.26	6 (9.2)	4 (6.2)	6 (9.2)	4 (6.2)	5 (7.7)	4 (6.2)
MLR < 0.24	PLR ≥ 143.26	3 (4.6)	1 (1.5)	3 (4.6)	1 (1.5)	2 (3.1)	3 (4.6)
	PLR < 143.26	4 (6.2)	15 (23.8)	4 (6.2)	14 (21.5)	3 (4.6)	15 (23.1)

DMARDs: disease-modifying anti-rheumatic drugs, RA: rheumatoid arthritis, NLR: neutrophil-to-lymphocyte ratio, PLR: platelet-to-lymphocyte ratio, MLR: monocyte-to-lymphocyte ratio.

## Data Availability

The data from this study are available on request to the corresponding author of the study.

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
