# Peer review of "Diagnostic Value of Neutrophil-to-Lymphocyte, Platelet-to-Lymphocyte, and Monocyte-to-Lymphocyte Ratios for the Assessment of Rheumatoid Arthritis in Patients with Undifferentiated Inflammatory Arthritis"

_diagnostics, 2022, doi:10.3390/diagnostics12071702_

Round 1
Reviewer 1 Report
This article tries to determine the utility of NLR, MLR, and PLR in the diagnosis of rheumatoid arthritis (RA) among subjects with undifferentiated inflammatory arthritis (UIA). Some concerns are enlisted below
1. How to choose the control subjects? Major systemic diseases (DM, CKD...) could affect the levels of NLR, MLR, and PLR. Do the authors consider propensity score matching ?
2. Line 189: " K-W test Kruskal-Wallis test," Typos should be corrected.
3. Please make it clear when describing the results of table 3 and 4 in terms of comparison of three types of RA to HEALTHY CONTROL and UIA.
Author Response
1. How to choose the control subjects? Major systemic diseases (DM, CKD...) could affect the levels of NLR, MLR, and PLR. Do the authors consider propensity score matching ?
: Healthy controls were randomly selected from female patients who visited the health promotion center of the same hospital for comprehensive routine health check-ups and had no history of rheumatologic disease, hematologic disease, malignancy, active infection, thrombotic disorders and any joint diseases. Neutrophil, lymphocyte, platelet, monocyte counts are greatly affected by inflammatory diseases, infection, hematologic disorders and malignancies. But we conjecture that the effect of DM, hypertension and CKD on NLR, PLR and MLR is minimal, we did not exclude DM, hypertension and CKD. Because the primary interest of the present study is not to compare NLR, PLR and MLR between controls and UIA patents, we do not consider propensity score matching
2. Line 189: " K-W test Kruskal-Wallis test," Typos should be corrected.
: “K-W test” is an abbreviation for “Kruskal-Wallis test”. Please review this comment again.
3. Please make it clear when describing the results of table 3 and 4 in terms of comparison of three types of RA to HEALTHY CONTROL and UIA.
: The results of Table 3 and 4 do not include healthy controls. The results of table 3 and 4 are analysis of data according to the presence or absence of 3 types of RA in patients with UIA. Table headings and their corresponding body text have been corrected for clarity in the revised manuscript.
Reviewer 2 Report
Many thanks to the authors for having presented a so interesting study about “Diagnostic value of neutrophil-to-lymphocyte ratio, platelet-to-lymphocyte ratio and monocyte-to-lymphocyte ratio for the assessment of rheumatoid arthritis in patients with undifferentiated inflammatory arthritis”. Please before resubmitting the revision version of the article, read the editorial rules carefully, and check for other editorial aspects, such as: text alignment, text justification at the head, etc. Please number the pages correctly. The language is so good that the manuscript does not need to be corrected by a person of English mother tongue.
Abstract
The abstract is well structured, and it contains the main results of the study.
Background
The introduction is quite well structured, containing the main aims of the study. The manuscript reflects the Strobe Statement-Checklist for this type of study. It should be noted that only female patients were analyzed in this study. Please discuss this aspect in the Discussion section.
Please, about OA quote also:
· Conditioned media from human osteoarthritic synovium induces inflammation in a synoviocyte cell line. Connect Tissue Res. 2019 Mar;60(2):136-145. doi: 10.1080/03008207.2018.1470167. Epub 2018 May 8. PMID: 29695173
Methods
This section contains enough information to understand and possibly repeat the study. In particular, it is well structured in relation to selection patients. Please explain the definition of undifferentiated inflammatory arthritis better: UIA should be regarded as a diagnosis of exclusion, so it would be appropriate to mention the major classes of disease to consider for the differential diagnosis. Please specify who performed statistical analysis: an independent statistician or the same authors?
Results
The results presented are quite complete, reflecting the MM section. Have you noticed differences in NLR, PLR and MLR in relation to the age of the patients?
Discussion
The length and content of the discussion communicate the main information of the paper. The most important limitation to this study is the one measurement of NLR, PLR, and MLR at the index day: it is difficult to evaluate the real diagnostic accuracy of these markers, which also depend on numerous variables.
Conclusion
The conclusions reflect and refer to the results of the study. There are some bias in this study: it should be implemented by adding the comparison with the male counterpart and increasing the measurements of the biomarkers under study, in order to make them more reliable diagnostic indicators of RA in patients with UIA.
References
The references are up to date; please integrate with that suggested previously.
Tables and Figures
The number and quality of tables are appropriate to transmit the main information of the paper.
Author Response
Abstract
The abstract is well structured, and it contains the main results of the study.
: Thank you for your kind comment.
Background
The introduction is quite well structured, containing the main aims of the study. The manuscript reflects the Strobe Statement-Checklist for this type of study. It should be noted that only female patients were analyzed in this study. Please discuss this aspect in the Discussion section.
: Thank you for your kind comment. We corrected the last sentence in the Introduction section to make it clear that this study only included female subjects. We also discuss this issue in the discussion section.
Please, about OA quote also:
- Conditioned media from human osteoarthritic synovium induces inflammation in a synoviocyte cell line. Connect Tissue Res. 2019 Mar;60(2):136-145. doi: 10.1080/03008207.2018.1470167. Epub 2018 May 8. PMID: 29695173
: We quote this reference in the introduction section.
Methods
This section contains enough information to understand and possibly repeat the study. In particular, it is well structured in relation to selection patients. Please explain the definition of undifferentiated inflammatory arthritis better: UIA should be regarded as a diagnosis of exclusion, so it would be appropriate to mention the major classes of disease to consider for the differential diagnosis. Please specify who performed statistical analysis: an independent statistician or the same authors?
: Major classes of disease to consider for the differential diagnosis for undifferentiated inflammatory arthritis includes rheumatoid arthritis as well as osteoarthritis, spondyloarthritis, crystal arthropathy and connective tissue diseases such as systemic lupus erythematosus etc. We added this notion in the method section as you suggest. The final diagnosis of undifferentiated inflammatory arthritis not classified as RA was as follows: Adult onset Still’s disease, n = 1; axial spondyloarthritis, n = 10; Behcet’s disease, n = 5; fibromyalgia n = 1; gout, n = 1; mixed connective tissue disease, n = 1; osteoarthritis, n = 80~83 (according to the type of RA such as clinical, classified and DMARDs RA), polymyalgia rheumatica, n = 3; unspecified arthritis, n = 17~21 (according to the type of RA such as clinical, classified and DMARDs RA); septic arthritis, n = 1; systemic lupus erythematosus, n = 5; Sjogren syndrome, n = 2; systemic sclerosis, n = 3; Takayasus’ arteritis, n = 1 and undifferentiated connective tissue disease, n = 1.
Aran Kim, co-author of this paper, performed statistical analysis. An explanation of who performed the statistical analysis was added to the manuscript.
Results
The results presented are quite complete, reflecting the MM section. Have you noticed differences in NLR, PLR and MLR in relation to the age of the patients?
: Age of UIA patients did not significantly correlate with NLR (ρ=0.001, p=0.993), PLR (ρ=0.044, p=0.534), and MLR (ρ=-0.112, p=0.114).
Discussion
The length and content of the discussion communicate the main information of the paper. The most important limitation to this study is the one measurement of NLR, PLR, and MLR at the index day: it is difficult to evaluate the real diagnostic accuracy of these markers, which also depend on numerous variables.
: We agree with your comment. As mentioned in discussion section, we think that this is one of the limitations of our study. We have added this to the discussion section of revised version of manuscript.
Conclusion
The conclusions reflect and refer to the results of the study. There are some bias in this study: it should be implemented by adding the comparison with the male counterpart and increasing the measurements of the biomarkers under study, in order to make them more reliable diagnostic indicators of RA in patients with UIA.
: As you comment, this is the one of the most important limitations of the present study. As we mentioned in the method and discussion section, due to the limited number of male patients with UIA in our center, we only analyzed female UIA patients. Because a previous study reported that NLR levels were higher in male patients with systemic autoimmune rheumatic diseases than in their female counterparts, we considered that the selection of only one gender group is appropriate for evaluating the clinical significance of NLR, PLR, and MLR. Thus, we think that further research is obviously necessary to overcome this limitation.
References
The references are up to date; please integrate with that suggested previously.
: We added more references in the revised manuscript, as you suggest.
Round 2
Reviewer 1 Report
The authors conjecture that the effect of DM, hypertension and CKD on NLR, PLR and MLR is minimal, so they did not exclude DM, hypertension and CKD. If there is any reference supporting the conjecture, please cite the reference.
Author Response
Thank you for your kind comment. Although there are data indicating that NLR, PLR, and MLR correlate with blood glucose level or are associated with the risk of diabetic complications such as cardiovascular diseases in patients with DM (eg. J Pak Med Assoc 2022 Jun;72(6):1097-1100. Neutrophil lymphocyte ratio as useful predictive tool for glycaemic control in type 2 diabetes: Retrospective, single centre study in Turkey; Ann Palliat Med 2022 Mar;11(3):984-992. The relationship between peripheral blood inflammatory markers and diabetic macular edema in patients with severe diabetic retinopathy), studies that analyze the differences in NLR, PLR, and MLR between DM and healthy controls are lacking. Similarly, rather than studying the relationship between hypertension and NLR, PLR, and MLR, studies on whether these markers are related to complications or outcome of hypertension have been published (eg. Risk Manag Healthc Policy 2022 Mar 9;15:427-433. The Relationship Between the Neutrophil to Lymphocyte Ratio, The Platelet to Lymphocyte Ratio, and Cardiac Syndrome X). Research on NLR, PLR, and MLR in patients with CKD mainly focuses on whether this indicator is related to the prognosis of CKD (eg. Biomedicines 2022 May 29;10:1272. The Predictive Value of NLR, MLR, and PLR in the Outcome of End-Stage Kidney Disease Patients).
Taken together, although we are not able to find any reference indicating that DM, hypertension, CKD do not affect the level of NLR, PLR and MLR level, we presumed that the effect of DM, hypertension and CKD on NLR, PLR and MLR is minimal in this study.
Reviewer 2 Report
The authors answered to my comments properly.
Well done!
Author Response
Thank you for your kind consideraiton.